# Unleashing the Multilingual Encoder Potential: Boosting Zero-Shot Performance via Probability Calibration

**Ercong Nie**[1,2]    **Helmut Schmid**[1]    **Hinrich Schütze**[1,2]
[1]Center for Information and Language Processing (CIS), LMU Munich, Germany
[2] Munich Center for Machine Learning (MCML), Germany
nie@cis.lmu.de

## Abstract

Pretrained multilingual encoder models can directly perform zero-shot multilingual tasks or linguistic probing by reformulating the input examples into cloze-style prompts. This is accomplished by predicting the probabilities of the label words at the masked token position, without requiring any updates to the model parameters. However, the performance of this method is limited by the model's bias toward predicting label words which frequently occurred during the pretraining. These words typically receive high probabilities. To address this issue, we combine the models with *calibration* techniques which modify the probabilities of label words predicted by the models. We first validate the effectiveness of a proposed simple calibration method together with other existing techniques on monolingual encoders in both zero- and few-shot scenarios. We subsequently employ these calibration techniques on multilingual encoders, resulting in substantial performance improvements across a wide range of tasks[1].

## 1 Introduction

Prompt-based learning (Brown et al., 2020; Liu et al., 2021) has emerged as an important research area. Recent research demonstrates that multilingual encoder models are capable of accomplishing zero-shot cross-lingual learning (Zhao and Schütze, 2021; Huang et al., 2022) and linguistic probing (Shapiro et al., 2021; Hartmann et al., 2021) by using cloze-style prompts. These prompts consist of an input sample, a task-specific context and a mask token. The encoder model applies Masked Language Modeling (MLM) (Devlin et al., 2019) to generate predictions for the mask token using a selection of prescribed candidate tokens from the vocabulary. These predictions can be subsequently utilized for label classification or probing purposes.

For example, the sentiment analysis of assigning the product review "*Worked as stated!*" to class POS can be reformulated as: "*Worked as stated!* `All in all, it was [MASK]`." The model is requested to fill in the word "*good*" at the mask token position, which is mapped to the POS label.

However, earlier studies indicate that the output of masked token prediction is biased towards certain label words in the candidate token list (Weissweiler et al., 2022; Nie et al., 2023). This bias not only affects the predicted class probabilities (Holtzman et al., 2021; Ahuja et al., 2022), but also deteriorates the model's overall performance (Zhao et al., 2021; Lu et al., 2022). According to Weissweiler et al. (2022) and Zhao et al. (2021), label words with higher frequency in the pretraining corpus tend to be predicted with higher probabilities. Besides, the prompt context can also influence the degree of bias present in the masked token prediction.

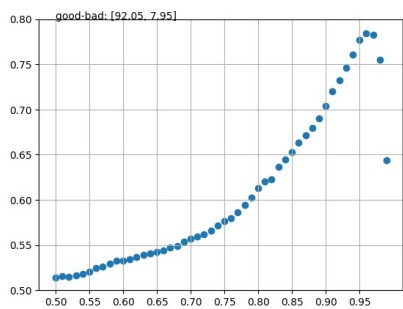

Figure 1: Example of the model predictions bias. The graph shows the accuracy on the amazon polarity test data (equally distributed) as a function of the classification threshold. $x$-axis refers to the threshold probability of good to classify examples with the class POS. The best results are obtained by classifying examples as POS if the probability of good exceeds 0.96.

Figure 1 illustrates the impact of the above mentioned biases on the model predictions. It shows the results of a binary sentiment analysis task with the BERT_Base (Devlin et al., 2019) model. The prompt template and label words used for this example can

---

[1]The code and data for this work are publicly available: https://github.com/ercong21/calibration.

| Method | Description | Probability Calculation | Source |
|--------|-------------|-------------------------|--------|
| CC | Contextual Calibration | $\tilde{q}(\mathbf{y}|x,t) = \mathbf{W}p(\mathbf{y}|x,t) + \mathbf{b}$ | Zhao et al. (2021) |
| PMI$_{DC}$ | Domain Conditional Pointwise Mutual Information | $\tilde{q}(\mathbf{y}|x,t) = log\frac{p(\mathbf{y}|x,t)}{p(\mathbf{y}|t)}$ | Holtzman et al. (2021) |
| CBM | Calibration By Marginalization | $\tilde{q}(\mathbf{y}|x,t) = \frac{p(\mathbf{y}|x,t)}{\frac{1}{|X|}\sum_{x' \in X} p(\mathbf{y}|x',t)}$ | Yang et al. (2023) |
| Penalty | Probability Penalty | $\tilde{q}(\mathbf{y}|x,t) = p(\mathbf{y}|x,t) + \mathbf{p}$ | Our proposed method |

Table 1: Overview of Calibration Methods. $\mathbf{y}$ refers to the label words. $X$ is the test dataset, $x$ is an input sample, and $t$ is the prompt template.

be found in Table 6. By shifting the threshold for predicting POS from 0.5 to approx. 0.95, the performance can be improved by more than 25%. Given only a mask token as input, the model predicts 0.92 and 0.08 as probabilities for the label words good and bad, respectively. To tackle the bias in the distribution of label words, our proposed solution in this work is to combine pretrained encoder models with *calibration* methods.

In this paper, we contribute by (1) proposing a simple yet effective calibration method that involves adding trainable penalties to the probabilities of the label words, (2) demonstrating its effectiveness in achieving performance enhancements comparable to other existing calibration techniques, (3) refining the calibration parameters with only a few training examples for further improvement, and (4) boosting the zero-shot performance of multilingual encoders by introducing calibration methods.

## 2 Calibration Methods

### 2.1 Existing Calibration Methods

**Contextual Calibration (CC)** Zhao et al. (2021) apply an affine transformation (Platt et al., 1999) to the original probabilities, as the first equation in Table 1 shows. The parameters of the affine transformation are obtained from the output probability distribution of the content-free input, e.g., the mask token, denoted $\hat{\mathbf{p}}_{cf}$. $\mathbf{W} = \text{diag}(\hat{\mathbf{p}}_{cf})^{-1}$ is the inverse diagonal matrix of $\hat{\mathbf{p}}_{cf}$ and $\mathbf{b}$ is an all-zero vector.

**Domain Conditional Pointwise Mutual Information (PMI$_{DC}$)** Holtzman et al. (2021) adjust the conditional class probability $p(\mathbf{y}|x,t)$ by dividing it with the prior probability $p(\mathbf{y}|t)$ of that class. We estimate $p(\mathbf{y}|t)$ for a given template $t$ using MLM with a prompt created by instantiating the prompt template with an empty input.

**Calibration By Marginalization (CBM)** Yang et al. (2023) are inspired by PMI$_{DC}$. Unlike PMI$_{DC}$, CBM approximates $p(\mathbf{y}|x,t)$ in a more precise manner by computing its marginalized probability, as the third equation in Table 1 shows. For each prediction, the sum probability $\Sigma_{x' \in X} p(\mathbf{y}|x',t)$ are calculated by taking all test inputs into account.

### 2.2 Our Method: Probability Penalty

Motivated by the observation in Figure 1 that a simple shift in the model's output distribution can substantially alleviate the label bias, we propose a penalty-based calibration approach as the equation in the last row of Table 1 shows. The core idea is to introduce a penalty term that is added to each individual label word probability. We initialize the corresponding parameter vector $\mathbf{p}$ with the negative prior probabilities of the label words. We estimate these prior probabilities using the output distribution of MLM applied to a mask token as input.

## 3 Experimental Setup

**Dataset** We first validate the effectiveness of the different calibration methods on several monolingual tasks. We study sentiment analysis using two datasets: binary **Amazon Polarity** (McAuley and Leskovec, 2013) and the English subset of 5-label **Multilingual Amazon Reviews** (Keung et al., 2020), topic categorization using two datasets: the **Ag News** and **Yahoo Answers Topics** (Zhang et al., 2015), sentence pair classification using two datasets: English subsets of **MNLI** (Conneau et al., 2018) and **PAWS-X** (Yang et al., 2019), and 5 datasets from the GLUE benchmark (Wang et al., 2019): **CoLA** (Warstadt et al., 2019), **MRPC** (Dolan and Brockett, 2005), **QQP**, **RTE** (Dagan et al., 2005), and **WNLI** (Levesque et al., 2012). For the evaluation of multilingual encoders, we use **Multilingual Amazon Reviews**, **XNLI** and **PAWS-X**. Besides, following Nie et al.

| | Balanced datasets (Acc.) | | | | | Imbalanced datasets (F1 Score) | | | | | | Avg. |
|---|---|---|---|---|---|---|---|---|---|---|---|---|
| | AG News | Amazon-P | Amazon-S | XNLI | Yahoo | Pawsx | CoLA | MRPC | QQP | RTE | WNLI | |
| BERT$_{Base}$ | | | | | | | | | | | | |
| + *no calib.* | 60.2 | 54.6 | 24.8 | 41.3 | 36.0 | 31.2 | 41.2 | 46.1 | 26.9 | 39.5 | 29.0 | 39.2 |
| + *CC* | **74.6** | 61.7 | 27.4 | 41.4 | 36.2 | 31.6 | 51.1 | 46.1 | 26.9 | 39.5 | **43.1** | 43.6 |
| + *PMI$_{DC}$* | 62.1 | 70.8 | 29.9 | 37.9 | 32.1 | 33.8 | **51.3** | 44.3 | 49.5 | 38.2 | 30.4 | 43.7 |
| + *CBM* | 73.6 | **71.3** | **33.6** | **42.9** | **45.2** | **49.3** | 49.9 | **50.6** | 52.6 | 50.9 | 42.3 | **51.1** |
| + *Penalty* | 67.9 | 61.7 | 26.3 | 42.6 | 39.4 | 31.6 | 51.1 | 46.1 | 26.9 | 39.5 | **43.1** | 43.3 |
| RoBERTa$_{Base}$ | | | | | | | | | | | | |
| + *no calib.* | 76.2 | 66.1 | 24.3 | 44.0 | 32.4 | 31.2 | 39.6 | 45.3 | 26.9 | 37.1 | 31.6 | 41.3 |
| + *CC* | 74.1 | **79.5** | 20.0 | 39.8 | 15.2 | 33.7 | 23.6 | 46.6 | 39.8 | 35.9 | 32.1 | 40.0 |
| + *PMI$_{DC}$* | 62.3 | 79.4 | **34.2** | 45.6 | 25.3 | 43.3 | 43.3 | **49.4** | 27.1 | 37.0 | 30.4 | 43.4 |
| + *CBM* | **78.4** | 76.5 | 34.1 | **46.4** | **42.9** | 44.4 | 48.2 | 47.5 | 50.1 | 43.3 | 49.0 | 51.0 |
| + *Penalty* | 75.6 | **79.5** | 30.1 | 41.4 | 26.9 | 33.7 | 23.6 | 46.6 | 39.8 | 35.9 | 32.1 | 42.3 |

Table 2: Results of zero-shot calibration methods on monolingual tasks. Amazon-P refers to Amazon Polarity (binary classification). Amazon-S refers to Amazon Star (5-way classification).

| BERT$_{Base}$ | | AG News | | Amazon-P | | Pawsx | | XNLI | | Avg | |
|---|---|---|---|---|---|---|---|---|---|---|---|
| nli-based ZR | | 54.9 | | **82.3** | | 48.2 | | 34.8 | | 55.1 | |
| calibration | | Penalty | CC | Penalty | CC | Penalty | CC | Penalty | CC | Penalty | CC |
| zero-shot | 0 | 67.9 | 74.6 | 61.7 | 61.7 | 45.4 | 45.4 | 42.6 | 41.4 | 54.4 | 55.8 |
| few-shot | 1 | 65.6$_{3.8}$ | 75.7$_{1.0}$ | 67.8$_{7.6}$ | 71.0$_{5.6}$ | 51.1$_{0.9}$ | 51.4$_{0.9}$ | 42.0$_{1.8}$ | 41.2$_{1.9}$ | 56.6$_{3.5}$ | 59.8$_{2.4}$ |
| | 2 | 67.2$_{3.1}$ | 75.9$_{1.6}$ | 71.9$_{4.4}$ | 72.2$_{3.2}$ | 51.0$_{1.1}$ | 50.7$_{1.0}$ | 42.7$_{0.6}$ | 42.5$_{0.9}$ | 58.2$_{2.3}$ | **60.3$_{1.7}$** |
| | 4 | 67.9$_{3.9}$ | 76.6$_{0.7}$ | 73.4$_{3.8}$ | 70.3$_{2.9}$ | **51.6$_{1.3}$** | 50.9$_{1.3}$ | 42.8$_{0.6}$ | 42.8$_{0.3}$ | 58.9$_{2.4}$ | 60.2$_{1.3}$ |
| | 8 | 69.1$_{1.5}$ | 76.9$_{0.1}$ | 75.2$_{2.3}$ | 71.8$_{1.2}$ | **51.6$_{1.1}$** | 49.9$_{0.6}$ | **42.9$_{0.2}$** | 42.7$_{0.2}$ | 59.7$_{1.3}$ | **60.3$_{0.5}$** |
| | 16 | 69.6$_{1.7}$ | **76.9$_{0.1}$** | 76.0$_{1.0}$ | 71.4$_{1.2}$ | 51.4$_{1.1}$ | 49.7$_{1.0}$ | 42.8$_{0.3}$ | 42.6$_{0.2}$ | 60.0$_{1.0}$ | 60.2$_{0.6}$ |
| RoBERTa$_{Base}$ | | AG News | | Amazon-P | | Pawsx | | XNLI | | Avg | |
| nli-based ZR | | 67.9 | | 84.8 | | 45.3 | | 34.3 | | 58.1 | |
| calibration | | Penalty | CC | Penalty | CC | Penalty | CC | Penalty | CC | Penalty | CC |
| zero-shot | 0 | 75.6 | 74.1 | 79.5 | 79.5 | 45.4 | 45.4 | 41.4 | 39.8 | 60.5 | 59.7 |
| few-shot | 1 | 75.6$_{2.6}$ | 77.2$_{1.5}$ | 77.4$_{8.0}$ | 81.3$_{4.9}$ | 48.4$_{1.8}$ | 48.4$_{1.4}$ | 45.9$_{0.9}$ | 44.8$_{1.5}$ | 61.8$_{3.3}$ | 62.9$_{2.3}$ |
| | 2 | 73.9$_{2.8}$ | 77.3$_{1.2}$ | 81.6$_{4.3}$ | 80.8$_{2.4}$ | 49.0$_{1.6}$ | 48.3$_{0.9}$ | 46.3$_{0.7}$ | 45.8$_{0.7}$ | 62.7$_{2.4}$ | 63.1$_{1.3}$ |
| | 4 | 74.5$_{1.9}$ | 77.6$_{1.0}$ | 82.2$_{4.4}$ | 79.6$_{1.6}$ | 49.3$_{0.6}$ | 48.5$_{0.9}$ | **47.2$_{0.2}$** | 46.0$_{0.3}$ | 63.3$_{1.8}$ | 62.9$_{1.0}$ |
| | 8 | 76.6$_{1.1}$ | 78.1$_{0.5}$ | **85.2$_{1.0}$** | 79.7$_{1.5}$ | **49.6$_{0.4}$** | 48.1$_{0.7}$ | 47.1$_{0.3}$ | 46.0$_{1.0}$ | 64.6$_{0.7}$ | 63.0$_{0.9}$ |
| | 16 | 78.3$_{0.5}$ | **78.4$_{0.3}$** | 85.1$_{1.0}$ | 79.7$_{1.6}$ | 49.4$_{0.6}$ | 48.1$_{0.4}$ | 47.0$_{0.2}$ | 46.0$_{0.9}$ | **65.0$_{0.6}$** | 63.1$_{0.8}$ |

Table 3: Results of few-shot calibration methods on monolingual tasks. *nli-based ZR* refers to the NLI-based zero-shot classification baseline (Yin et al., 2019).

(2023), we expand the **AG News** dataset to 25 languages using machine translation to conduct a wide range of cross-lingual analyses.

**Setup** In our monolingual experiments, we use the pretrained models `bert-base-cased` (Devlin et al., 2019) and `roberta-base` (Liu et al., 2019). In the multilingual experiments, we use their multilingual counterparts `bert-base-multilingual-cased` and `xlm-roberta-base` (Conneau et al., 2020). We use PyTorch (Paszke et al., 2019) and the HuggingFace framework (Wolf et al., 2020). We repeat each experiment 5 times with different random seeds and report the mean and variance. Details of the experimental setting can be found in Appendix A.

# 4 Results and Analysis

## 4.1 Results on Monolingual Encoders

### 4.1.1 Zero-shot calibration

We first validate the effectiveness of the various calibration methods on monolingual encoders. Table 2 shows the results of zero-shot calibration, where we directly calculate the calibrated probabilities without using additional training samples. We report accuracies for evenly distributed datasets and F1 scores for imbalanced datasets. Compared to the uncalibrated baseline systems, we obtain improvements across most of the tasks, except for the *CC* method combined with the RoBERTa model. In this specific case, the average performance worsens compared to the *no calibration* baseline due to outlier performance observed in several tasks, such as Yahoo and CoLA.

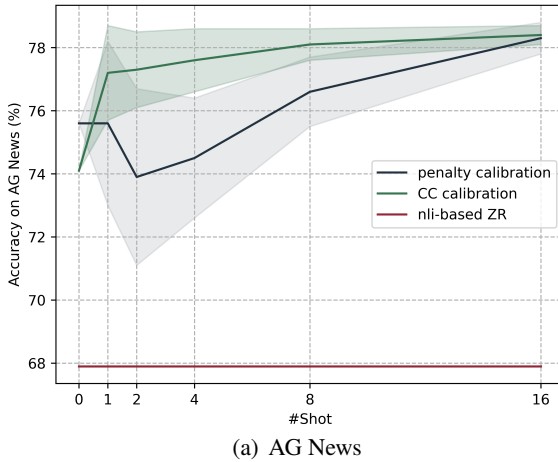

(a) AG News

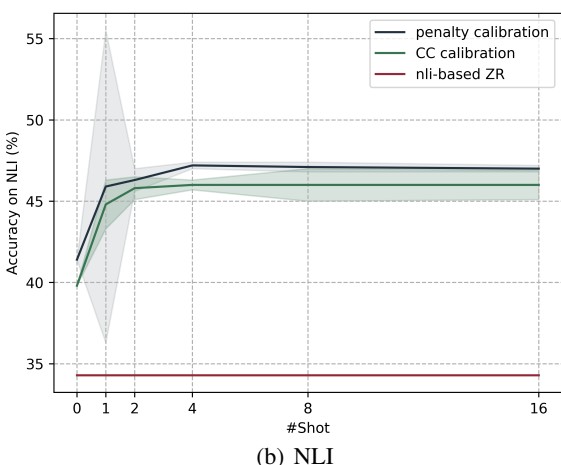

(b) NLI

Figure 2: Performance and variation of few-shot calibration on the RoBERTa model.

### 4.1.2 Adding few-shot samples further boosts the performance

As the formulas in Table 1 show, $PMI_{DC}$ and $CBM$ directly modify the probabilities without introducing additional parameters, while $CC$ and $Penalty$ use specific calibration parameters, which are trainable. In zero-shot calibration, these parameters are initialized by prior probabilities without being updated. We will now make use of the trainability of parameters in $CC$ and $Penalty$ to investigate if applying few-shot training to calibration parameters further improves the performance.

Table 3 shows the results of few-shot calibration. We observe that training the calibration parameters on just a few samples further enhances the performance of the calibrated systems. Compared to zero-shot calibration, few-shot calibration achieves better performance in most cases. We also compare calibration methods in few-shot scenarios with the NLI-based zero-shot classification base-

line proposed by Yin et al. (2019). Details of the baseline setting and the few-shot training process are described in Appendices A.3 and B.

Figure 2 shows the few-shot calibration results of the RoBERTa model on the AG News and NLI tasks. Prior research (Zhao and Schütze, 2021) showed that few-shot learning can be unstable due to the randomness. However, as Figure 2 shows, the variation in performance diminishes obviously as the number of shots increases. Our experimental results indicate that few-shot calibration not only enhances the performance but also increases the steadiness.

### 4.2 Results on Multilingual Encoders

Table 4 shows our experimental results on multilingual datasets, indicating that calibration methods are also effective for multilingual encoders.

Our experiments cover a large range of languages considering both language availability, i.e., if or how much language data exists in the pretraining corpus, and language diversity, i.e., to which language family a language belongs. Specifically, for Amazon-S, XNLI and PAWS-X, we use the original test sets, mainly containing high-resource languages. In the multilingual AG News task, we include many low-resource and unseen languages by generating parallel multilingual test sets using machine translation techniques. Recent research by Hu et al. (2020) and Liu et al. (2022) shows that automatically translated test sets are useful for measuring cross-lingual performance. Hence, we adopt their methodology and expand the language coverage of the AG News dataset to 25. The list of languages can be found in Appendix C.

The results on multilingual BERT and XLM-R show that all four calibration methods improve the multilingual performance averaged across all tasks. For both models, $CBM$ always emerges as the top-performing approach. Different from other approaches predicting the label with one input by another, $CBM$ is the only method which leverages the test set (without labels) to adjust the calibration parameters. This could account for the substantial advantage of $CBM$ over the others in terms of the performance.

### 4.3 Multilingual Analysis

Now we analyze how different language properties correlate with the performance of multilingual BERT on the AG News task.

|  | AG News | Amazon-S | XNLI | PAWS-X | Avg. |
|---|---|---|---|---|---|
| mBERT$_{Base}$ |  |  |  |  |  |
| + *no calib.* | 32.8 | 20.5 | 33.6 | 33.9 | 30.2 |
| + *PMI$_{DC}$* | 48.8 | 22.5 | 33.6 | 44.4 | 37.3 |
| + *CBM* | 53.8 | **25.1** | 34.9 | **49.2** | **40.8** |
| + *CC (max)* | 53.9 | 23.9 | 35.1 | 44.8 | 39.4 |
| + *Penalty (max)* | **54.6** | 23.8 | **35.3** | 47.1 | 40.2 |
| XLM-R$_{Base}$ |  |  |  |  |  |
| + *no calib.* | 45.4 | 21.9 | 35.0 | 31.7 | 33.5 |
| + *PMI$_{DC}$* | 59.8 | 23.0 | 33.6 | 37.8 | 38.6 |
| + *CBM* | **63.3** | **28.9** | **37.8** | **46.3** | **44.1** |
| + *CC (max)* | 59.6 | 23.7 | 35.3 | 43.7 | 40.6 |
| + *Penalty (max)* | 57.5 | 23.6 | 35.8 | 43.4 | 40.1 |

Table 4: Results of calibration methods on multilingual datasets. We report the best results for *CC* and *Penalty* in different few-shot settings.

### 4.3.1 Language Accessibility

We first group the evaluation languages into low-resource languages, unseen languages, and languages with unseen scripts to determine the influence of language accessibility. Low-resource languages are languages which are contained in the pretraining corpus, but only account for a small amount of it. Unseen languages do not occur in the pretraining, thus the multilingual encoder has never seen them. The hardest case involves languages with unseen scripts, where the model has not even encountered the characters of the language. However, our test set contains no languages with completely unseen scripts because machine translation frequently generates code-switched data. Figure 3 (a) shows that low-resource languages perform generally better than the other two types of unseen languages, indicating that the multilingual encoder's access to languages in the pretraining is crucial for the performance enhancement via calibration.

### 4.3.2 Language Diversity

We further group the languages according to their phylogenetic relationships, i.e., from which language family they are. We analyze the language families containing at least 3 languages. The box plots in Figure 3 (b) reveal that the impact of calibrating multilingual encoders varies across different language groups. Specifically, we observe that Indo-European and Dravidian languages tend to benefit more from calibration than Austronesian and Niger-Congo languages.

This discrepancy suggests that the effectiveness of calibration techniques can be influenced by the language accessibility of multilingual encoders and the linguistic characteristics of language families.

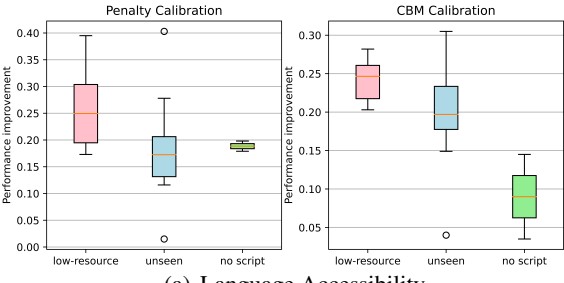

(a) Language Accessibility

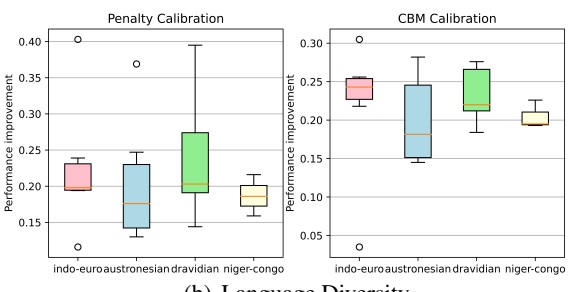

(b) Language Diversity

Figure 3: Performance Improvement of multilingual BERT with two calibration methods.

## 5 Conclusion

In conclusion, our work focuses on boosting the zero-shot learning performance of multilingual encoders in language understanding tasks through probability calibration techniques. We address the bias issue in the mask token prediction of label words by introducing various calibration techniques that modify the probabilities of these words. We first test the efficacy of different calibration methods in monolingual encoders. We also prove that with a minimal number of training examples, the calibrated probabilities yield further enhancements compared to the zero-shot calibration method. Our experiments on multilingual encoders demonstrate that all calibration methods bring a performance improvement across various tasks.

## Limitations

We propose a simple yet effective calibration method to enhance the zero-shot performance for monolingual and multilingual encoders. While our work shows the effectiveness of calibration for enhancing the prediction with multilingual tasks, it is important to note that our research is primarily focused on classification tasks with multilingual encoders. As a result, our findings and proposed methods may not directly translate to generation tasks, such as question answering (QA), which involve the use of generative multilingual models. Future investigations should explore the application of our calibration methods on generation tasks and evaluate their effectiveness in enhancing the performance of generative multilingual models. This extension could provide valuable insights into the potential benefits and limitations of our approaches across a broader range of NLP tasks.

## Ethics Statement

This research was conducted in accordance with the ACM Code of Ethics. All the datasets that we use are publicly available. We report only aggregated results in the main paper. We do not share any Personally Identifiable Data in this paper.

## Acknowledgements

We extend our sincere gratitude to the anonymous reviewers for their invaluable contributions and constructive feedback that have greatly enriched the quality and scope of this paper. This work was supported by Munich Center for Machine Learning (MCML) and China Scholarship Council (CSC).

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

## A    Experimental Details

This section provides a comprehensive overview of our experimental setup, including hyperparameters, prompt templates that we use in our experiments, and the baselines.

### A.1    Hyperparameters

To ensure experimental reproducibility, we present the hyperparameter settings used in our study in Table 5.

| Hyperparameter | Value |
|---|---|
| Evaluation batch size | 8 |
| Learning rate | 1e-4 |
| Random seeds | {42, 421, 512, 1213, 1234} |
| Maximal sequence length | 128 |
| Few-shot numbers | {1, 2, 4, 8, 16} |
| GPU type | NVIDIA GeForce GTX 1080 Ti |
| Number of GPU | 8 |

Table 5: Overview of hyperparameters.

### A.2    Prompt Engineering

We select a set of prompt templates for the tasks through our preliminary experiments. Table 6 shows the prompt templates and the label words used in our experiment.

### A.3    Baseline

To establish a baseline, we initially conduct experiments without employing any calibration methods. Subsequently, we introduce four calibration methods individually and evaluate their impact on the performance. Besides, we compare our calibration methods with an NLI-based zero-shot classification baseline proposed by Yin et al. (2019), where they first finetune a pretrained language model on the MNLI dataset, then they reformulate common classification tasks to an NLI task format. The input sample is regarded as the premise, while the label serves as the hypothesis. The zero-shot classification is performed by directly comparing the probabilities of predicting `entailment` for all input-label pairs. For this baseline, we finetune a BERT model and a RoBERTa model on the MNLI task.

## B    Few-Shot Training of Calibration Parameters

Algorithm 1 presents the process of few-shot training of penalty calibration used in our few-shot investigation.

---

**Algorithm 1:** Few-Shot Training of Penalty Calibration

---

**Input:** set of few-shot training samples $D$, initial calibration parameter vector $p_0$, number of epochs $E$, learning rate $\eta$

**Output:** Trained parameters $p$

Initialize $p \leftarrow p_0$;
**for** *epoch* ***in*** $1, 2, \cdots, E$ **do**
 **foreach** $(x, y)$ ***in*** $D$ **do**
  $l \leftarrow get\_probs(x)$;
  $l \leftarrow l - p$ # calibration;
  $\hat{y} \leftarrow argmax_y(l[y])$;
  **if** $y \neq \hat{y}$ **then**
   $p[\hat{y}] \leftarrow p[\hat{y}] + \eta$;
   $p[y] \leftarrow p[y] - \eta$;
  **end**
 **end**
**end**

---

## C    Detailed Results

Detailed results of the experiments in the main text can be found in this section. Table 8 shows the complete results of mBERT on the multilingual AG News dataset across all 25 languages. Table 7 provides an overview of languages covered by the multilingual AG News dataset.

| Task | Prompt template | Label words |
|------|-----------------|-------------|
| Ag News | mask News: [X] | 'World', 'Sports', 'Business', 'Tech' |
| Amazon-P | [X]. All in all, it was mask. | 'bad', 'good' |
| Amazon-P | [X]. All in all, it was mask. | 'terrible', 'bad', 'ok', 'good', 'great' |
| XNLI | [X]? mask, [Y] | 'Yes', 'Maybe', 'No' |
| Yahoo | mask Question: [X] [Y] | 'Society', 'Science', 'Health', 'Education', · · · |
| PAWS-X | [X] . mask[ Y] | 'Wrong', 'Right' |
| CoLA | [X] . It is linguistially mask. | 'wrong', 'right' |
| MRPC | [X]? mask, [Y] | 'Wrong', 'Right' |
| QQP | Question 1: [X] Question 2: [Y] Question 1 and Question 2 are mask | 'different', 'same' |
| RTE | [X]? mask, [Y] | 'Wrong', 'Right' |
| WNLI | [X]? mask, [Y] | 'Wrong', 'Right' |

Table 6: Overview of prompt templates.

| Code | Languages | Language Accessibility | Language Family |
|------|-----------|------------------------|-----------------|
| af | Afrikaans | Low-resource | Indo-European |
| co | Corsican | Unseen languages | Indo-European |
| eo | Esperanto | Unseen languages | Artificial |
| haw | Hawaiian | Unseen languages | Austronesian |
| hmn | Hmong | Unseen languages | Sino-Tibetan |
| ht | Haitian Creole | Low-resource | Indo-European |
| ig | Igbo | Unseen languages | Niger-Congo |
| jw | Javanese | Low-resource | Austronesian |
| km | Khmer | Unseen script | Austronesian |
| mi | Maori | Low-resource | Austronesian |
| mn | Mongolian | Low-resource | mongolian |
| mt | Maltese | Unseen languages | Afro-Asiatic |
| my | Burmese | Low-resource | Sino-Tibetan |
| ny | Chichewa | Unseen languages | Niger-Congo |
| or | Odia | Unseen script | Indo-European |
| sm | Samoan | Unseen languages | Austronesian |
| sn | Shona | Unseen languages | Dravadian |
| st | Sesotho | Unseen languages | Dravadian |
| sw | Swahili | Low-resource | Dravadian |
| ta | Tagalog | Low-resource | Austronesian |
| te | Telugu | Low-resource | Dravadian |
| tl | Tamil | Low-resource | Dravadian |
| ug | Uighur | Unseen languages | Turkic |
| ur | Urdu | Low-resource | Indo-European |
| uz | Uzbek | Low-resource | Turkic |
| zu | Zulu | Unseen languages | Niger-Congo |

Table 7: Overview of languages covered by the multilingual AG News dataset.

| | af | co | en | eo | haw | hmn | ht | ig | jw | km | mi | mn | mt | my |
|---|----|----|----|----|-----|-----|----|----|----|----|----|----|----|----|
| No calib. | 40.4 | 32.6 | 47.3 | 31.9 | 27.1 | 30.9 | 35.7 | 30.2 | 38.0 | 33.3 | 29.0 | 32.0 | 29.9 | 33.8 |
| Penalty | 64.3 | 44.2 | 69.6 | 72.3 | 40.1 | 49.6 | 55.2 | 48.8 | 62.6 | 51.2 | 46.3 | 62.2 | 57.6 | 64.7 |
| CBM | 64.7 | 58.3 | 69.1 | 62.4 | 42.0 | 50.8 | 60.9 | 49.6 | 63.9 | 47.8 | 49.5 | 53.0 | 57.2 | 54.1 |
| CC | 65.6 | 59.7 | 67.8 | 68.0 | 43.4 | 49.7 | 65.2 | 52.4 | 66.4 | 41.4 | 51.2 | 55.4 | 57.4 | 51.7 |
| PMI$_{DC}$ | 60.2 | 35.3 | 60.0 | 61.7 | 35.9 | 33.5 | 33.5 | 49.2 | 61.5 | 42.2 | 49.6 | 54.7 | 61.1 | 47.6 |

| | ny | or | sm | sn | st | sw | ta | te | tl | ug | ur | uz | zu | avg. |
|---|----|----|----|----|----|----|----|----|----|----|----|----|----|------|
| No calib. | 29.8 | 25.4 | 30.3 | 32.2 | 30.4 | 33.4 | 28.8 | 32.5 | 42.6 | 25.5 | 33.2 | 33.9 | 34.5 | 32.8 |
| Penalty | 51.4 | 45.2 | 43.5 | 52.4 | 44.8 | 72.9 | 65.6 | 59.9 | 61.7 | 27.0 | 52.6 | 59.1 | 50.3 | 54.6 |
| CBM | 52.4 | 28.9 | 46.1 | 53.4 | 48.8 | 59.9 | 57.0 | 60.0 | 64.6 | 29.5 | 56.8 | 58.9 | 53.7 | 53.8 |
| CC | 51.2 | 28.7 | 47.5 | 52.5 | 49.1 | 64.1 | 56.5 | 52.4 | 62.6 | 27.9 | 53.1 | 60.3 | 49.6 | 53.7 |
| PMI$_{DC}$ | 50.2 | 28.6 | 43.9 | 50.9 | 44.6 | 61.6 | 50.1 | 43.6 | 66.1 | 29.3 | 55.0 | 56.4 | 51.3 | 48.8 |

Table 8: Results of mBERT on the multilingual AG News dataset across all languages.