# OpenReview forum: "Unleashing the Multilingual Encoder Potential: Boosting Zero-Shot Performance via Probability Calibration"
_EMNLP/2023/Conference — EMNLP 2023 Findings_

### Official Review · Reviewer_Pst8 · 2023-08-02

**Soundness:** 3

**Excitement:**

3: Ambivalent: It has merits (e.g., it reports state-of-the-art results, the idea is nice), but there are key weaknesses (e.g., it describes incremental work), and it can significantly benefit from another round of revision. However, I won't object to accepting it if my co-reviewers champion it.

**Paper Topic And Main Contributions:**

The paper focuses on enhancing zero-shot performance of pretrained encoders studying classification tasks. The method involves modifying the word probabilities of the model with some calibration method. The main novelty is the introduction of a new calibration method "penalty", which adds a penalty to each word. The approach is then tested on multiple datasets for both the monolingual and multilingual case.

**Questions For The Authors:**

A) In figure 2, should we also compare as a baseline the other zero-shot calibration methods? For example, CBM  zero-shot in AgNews does 78.4 which is better than the baseline and also CC and "penalty" few-shot.

B) In the figure 1 case what is the class distribution probability?

**Reasons To Accept:**

 - The paper introduces a new calibration method that often improves against the base model without any modifications.

- The experimental setup is solid with multiple datasets and other methods for comparison.

 - The paper is clear and well written,

**Reasons To Reject:**

- While the proposed approach shows improvement over the bare model, the results are not good when compared with other methods from the literature which are consistently outperforming it.

- The paper is framed (e.g. both title and abstract) on multilingual encoders. However in the setup both the monolingual and multilingual aspects are equally studied.

**Reproducibility:**

4: Could mostly reproduce the results, but there may be some variation because of sample variance or minor variations in their interpretation of the protocol or method.

**Reviewer Confidence:**

4: Quite sure. I tried to check the important points carefully. It's unlikely, though conceivable, that I missed something that should affect my ratings.

**Typos Grammar Style And Presentation Improvements:**

- The Colors of figure 2 can be differentiated a bit more, in particular AgNews

- The first word of the abstract is misspelled

---

> ### Author Rebuttal · Authors · 2023-08-28
>
> We extend our gratitude for your careful evaluation of our paper. Your acknowledgement of contributions we have presented is sincerely appreciated. We are glad to address your insightful comments and suggestions, particularly with regard to certain aspects where there might be some misunderstanding. Before elaborating on concrete points, I would like to first reiterate the contributions of our work:
> Our contributions encompass two distinct aspects.
> 1. Our work establishes the effectiveness of the proposed $Penalty$ calibration method:
> a. We introduce a straightforward yet potent calibration method.
> b. Our method demonstrates performance on par with existing calibration techniques.
> 2. Our contributions extend to a comprehensive examination of various calibration methods:
>      c.  Few-shot calibration basically outperforms zero-shot calibration.
>      d. Calibration methods work on multilingual encoders as well.
>
> **Response to Reasons to Reject**
> As pointed out in your **first** piece of reason, our proposed calibration method failed to outperform other calibration methods in many cases. However, as claimed in the Introduction section (line 073-078), the major contribution with regard to the proposed $penalty$ method is that we introduce a simple yet effective calibration method that can achieve **comparable** results to other existing techniques. We did not intend to invent a SOTA calibration method, but to offer more effective options that can be applied to few-shot and multilingual scenarios (the second contribution as mentioned above).
>
> As response to the **second** point stating that the setup of this study in both monolingual and multilingual aspects is not in line with the title and abstract framed on multilingual encoders, we would like to sort out our overall thoughts of this paper: The motivation of this paper is to boost the zero-shot learning performance of multilingual encoder models through probability calibration techniques, including both existing calibration techniques and one proposed calibration approach. For better using calibration techniques, we also proposed a few-shot calibration method: training calibration parameters based on very few training samples. Before implementing these techniques into multilingual encoders, we initially assessed their efficacy through experiments in monolingual settings. After that, we extended their application to multilingual encoders, effectively showcasing their utility in multilingual encoders.
>
> We would like to modify our writing to present our contributions and thinking in a more clear, more organised and more easily understandable way.
>
> **Response to Questions**
> **Q(a)**: As explained in line 171-178, the reason why we used CC and Penalty calibration methods for few-shot training but excluded $PMI_{DC}$ and $CBM$ is that $PMI_{DC}$ and $CBM$ directly calculate the calibrated probabilities using probabilistic formulas without using additional trainale parameters. Different from $PMI_{DC}$ and $CBM$, $CC$ and $Penalty$ contain trainable parameters. As Section 2 shows, the parameters for $CC$ and $Penalty$ can be initialised in the zero-shot scenario. Only calibration methods with trainable parameters fit the few-shot training methods.
> It was regrettable that due to the 4-page range limitation, we did not give a very explicit explanation to this point in the paper. If we have the chance to have our paper accepted, we would like to use the additional one page for the camera-ready version to add more detailed elaborations in the main text.
>
> **Q(b)**: Thanks for your question on the class distribution probability in Figure 1. It is 0.5:0.5. We presented our example on a balanced test set. It is our negligence not to mark it. We will add this information in the future version. Thanks for the reminder!
>
> **Response to Typos**:
> Thanks for your thorough review! We will promptly rectify these typo errors.
> Thanks for your advice on the colouring match of Figure 2. We will recolor it to make the lines more differentiated!
>
> Thank you once more for your comprehensive review. We sincerely wish that we delve into your concerns in a detailed manner.

---

### Official Review · Reviewer_LfMb · 2023-08-08

**Soundness:** 3

**Excitement:**

3: Ambivalent: It has merits (e.g., it reports state-of-the-art results, the idea is nice), but there are key weaknesses (e.g., it describes incremental work), and it can significantly benefit from another round of revision. However, I won't object to accepting it if my co-reviewers champion it.

**Paper Topic And Main Contributions:**

This paper proposes to combine the models with various calibration techniques which modify the probabilities of label words predicted by the models. The experimental results on several benchmarks verify the effectiveness of the proposed method.

**Reasons To Accept:**

(1) This paper shows the effectiveness of calibration techniques in the zero-shot scenario; (2) The experiments are sufficient and the results are comprehensive and convincing.

**Reasons To Reject:**

The novelty is limited. The use of calibration techniques is straight.

**Reproducibility:**

3: Could reproduce the results with some difficulty. The settings of parameters are underspecified or subjectively determined; the training/evaluation data are not widely available.

**Reviewer Confidence:**

4: Quite sure. I tried to check the important points carefully. It's unlikely, though conceivable, that I missed something that should affect my ratings.

---

> ### Author Rebuttal · Authors · 2023-08-28
>
> Thanks for your time and effort you have invested in providing feedback on our work. In order to avoid misleading, we would like to take the opportunity to emphasise the contributions of our work again:
>
> Our contributions can be summarised from two aspects:
> 1. We prove the efficacy of our proposed $Penalty$ calibration method:
> a. We proposed a simple but effective calibration method
> b. Our method can achieve comparable performance to the existing calibration techniques.
> 2. The next aspect of our work’s contributions treats different calibration methods as a whole:
> c. few-shot calibration performs better than zero-shot in principle.
> d. calibration methods take effect on multilingual encoders as well.
>
> Regarding the "use of calibration techniques" which you mentioned in the "Reasons to reject", as mentioned in the Limitation section, as a short paper, we focus on validating the effectiveness of the calibration techniques for multilingual encoders and in few-shot scenarios. We will continue to explore the calibration methods for autoregressive models and multilingual LLMs in our future work.

---

### Official Review · Reviewer_3ZFj · 2023-08-11

**Soundness:** 3

**Excitement:**

4: Strong: This paper deepens the understanding of some phenomenon or lowers the barriers to an existing research direction.

**Missing References:**

Although you discuss them in your limitations section, I believe calibration work in autoregressive LMs should be at least discussed briefly. One reference which I was surprised was not included is the following:
1. Min, Sewon, et al. "Noisy channel language model prompting for few-shot text classification." arXiv preprint arXiv:2108.04106 (2021).

**Paper Topic And Main Contributions:**

This paper considers the setting in which a the input to a task is reformulated as a masked language modeling task e.g., in the sentiment classification of a movie review converting the input and label ("This was a great movie", Positive) to a masked language modeling task "This was a great movie. The movie was [MASK]." The authors discuss that, although this task conversion is flexible, it often suffers from poorly calibrated logits being used to decide the replacement of the [MASK]. The authors then offer a calibration method which they test on both unilingua multilingual language modeling encoders and which essentially "shifts" the thresholding based on the uncalibrated logit score. The authors show that this simple calibration method allows unilingual and multilingual MLMs to perform better than uncalibrated ones.

**Reasons To Accept:**

I would like to take this opportunity to congratulate the authors on a nice piece of work.
- The work tackles an important, under-explored problem in the are of calibrating MLMs which has implications on model reuse and the use of task specific fintetuning.
- The method proposed is simple and intuitive and does better than no calibration in most cases and sometimes outperforms SOTA calibration methods. (e.g. Amazon-P for the unilingual case and AGNews and XNLI for the multilingual case)
- The experiments are thorough and surpass my expectations for a short paper.

**Reasons To Reject:**

I believe this paper is worth accepting but only if the primary concern discussed below is addressed.
- Primary concern: The results, analysis and conclusions of the paper do not line up with the claimed contributions. In particular, paragraph 73-82 claims that "its [the calibration method] effectiveness in achieving performance enhancements comparable to other existing calibration techniques" which is amplified in the conclusion where a claim is made that "the calibrated probabilities yield significant enhancements". I don't believe this to be entirely true as, in several cases, the "Penalty" method performs much worse than current methods. This point is made not to devalue the author's contributions nor to use a "Not SOTA so Reject" argument, but rather because several, if not the majority, of results seem to dispute the authors' claim. I would consider slightly reframing the paper to be more exploratory/case-study-like.

**Reproducibility:**

5: Could easily reproduce the results.

**Reviewer Confidence:**

5: Positive that my evaluation is correct. I read the paper very carefully and I am very familiar with related work.

**Typos Grammar Style And Presentation Improvements:**

- Line 001: Pretraiend
- Line 181: "after of"

---

> ### Author Rebuttal · Authors · 2023-08-28
>
> We sincerely appreciate your thoughtful review and valuable feedback on our paper. Your acknowledgement of our work serves as a source of encouragement to us. Subsequently, we would like to address a few points that were raised in your review, particularly regarding some aspects that might have led to certain misconceptions.
>
> **Response to your primary concern**
> You expressed your concern that “the results, analysis and conclusions of the paper do not line up with the claimed contributions” and advise us to “slightly reframe the paper to be more exploratory/case-study like”.
>
> Your insightful concern helped us realise that there might exist some unclarity and ambiguity in our writing. This could potentially lead to deviations from our intended meaning and create misunderstandings among our readers. We would like to take this opportunity to explain our thinking. In the conclusion part, by writing “Notably, with a minimal number of training examples, the calibrated probabilities yield significant enhancements”, we are trying to highlight the effectiveness of few-shot training of calibration, for both calibration methods, i.e. our proposed $Penalty$ and the existing $CC$. As Table 7 shows, most of the few-shot cases outperforms the zero-shot case across all experimented tasks, in many cases even with a substantial margin.
>
> We did not intend to amplify the effect of our proposed $Penalty$ calibration in our work. Our contributions (as outlined in Sec. 1) can be categorised into two aspects:
> 1. proving the effectiveness of the $Penalty$ calibration method whose performance is comparable to the existing methods
> 2. proving the general effectiveness across different calibration methods in multilingual encoders and in few-shot settings: few-shot calibration performs better than zero-shot in principle; calibration methods take effect on multilingual encoders as well.
>
> We appreciate the valuable concern you pointed out. To mitigate any potential misunderstandings, we would reorganise our language, ensuring that our contributions are conveyed with greater clarity and precision. Besides, as you suggested, reframing the paper to add more exploratory case analysis would contribute to enhancing its clarity. While we initially considered conducting a more in-depth case analysis, we regrettably had to forgo this due to the constraints in scope. However, if our paper is accepted, we would like to make use of the additional page for the camera-ready version to incoporate more case studies, aligning with your advice.
>
> **Response to Missing references**
> Thank you for pointing this out. We neglected the paper you referred to regarding the relevant work on autoregressive models. We will add this reference in our next version.
>
> **Response to Typos**
> Thanks for your careful reading! We will promptly fix these typos.
>
> Thank you once again for your thorough review, and we hope our explanations could help address your concerns.

---

### Meta-Review · Area_Chair_2SvR · 2023-09-19

**Recommendation:** 3

**Metareview:**

This paper propose a calibration method to mitigate the bias towards high confidence predictions. The experimental results of this paper show empirical evidence of substantial performance gains, even with just a few training samples. Overall this incremental, simple, yet effective method is a perfect fit for a shot paper. The main concern that the authors should address are the ones expressed by rev. 3ZFj on mismatch of the empirical results with some claims. It would also help to include the monolingual contributions into the title, as the multilingual focused title is misleading (rev. Pst8).

---

### Decision · Program_Chairs · 2023-10-07

**Decision:**

Accept-Findings

**Comment:**

This paper propose a calibration method to mitigate the bias towards high confidence predictions. The experimental results of this paper show empirical evidence of substantial performance gains, even with just a few training samples. Overall this incremental, simple, yet effective method is a perfect fit for a shot paper. The main concern that the authors should address are the ones expressed by rev. 3ZFj on mismatch of the empirical results with some claims. It would also help to include the monolingual contributions into the title, as the multilingual focused title is misleading (rev. Pst8).